# Reliable Methods for Classification, Characterization, and Design of Cellular Structures for Patient-Specific Implants

**DOI:** 10.3390/ma16114146

**Published:** 2023-06-02

**Authors:** István Nemes-Károly, Gábor Szebényi

**Affiliations:** 1Department of Polymer Engineering, Faculty of Mechanical Engineering, Budapest University of Technology and Economics, Műegyetem rkp. 3, H-1111 Budapest, Hungary; nemes-karolyi@pt.bme.hu; 2MTA-BME Lendület Lightweight Polymer Composites Research Group, Műegyetem rkp. 3, H-1111 Budapest, Hungary

**Keywords:** periodic cellular structures, classification, DMLS, mechanical properties, optical strain measurement, stiffness, FE simulation, drug storage, osseointegration, stress shielding

## Abstract

In our research, our goal was to develop a characterization method that can be universally applied to periodic cell structures. Our work involved the accurate tuning of the stiffness properties of cellular structure components that can significantly reduce the number of revision surgeries. Up to date porous, cellular structures provide the best possible osseointegration, while stress shielding and micromovements at the bone-implant interface can be reduced by implants with elastic properties equivalent to bone tissue. Furthermore, it is possible to store a drug inside implants with a cellular structure, for which we have also prepared a viable model. In the literature, there is currently no established uniform stiffness sizing procedure for periodic cellular structures but also no uniform designation to identify the structures. A uniform marking system for cellular structures was proposed. We developed a multi-step exact stiffness design and validation methodology. The method consists of a combination of FE (Finite Element) simulations and mechanical compression tests with fine strain measurement, which are finally used to accurately set the stiffness of components. We succeeded in reducing the stiffness of test specimens designed by us to a level equivalent to that of bone (7–30 GPa), and all of this was also validated with FE simulation.

## 1. Introduction

Cellular structures are gaining ground in a wide variety of industries, appearing and playing a dominant role in numerous devices and disciplines, from medical devices to filtration technology and automotive. They have a number of favorable properties that bulk materials do not have or have only to a limited extent, such as high stiffness per low density, mechanical properties that can be designed and adjusted according to requirements, favorable sound and thermal insulation properties, good vibration damping capacity, and outstanding energy absorption. As our research is mainly dealing with mechanical properties, including the elastic modulus, we will focus on these in more detail. The favorable mechanical properties of cellular structures are mainly used by the transportation industry, as they allow significant weight savings without reducing the stiffness of the components, thus saving a significant amount of raw material and fuel while increasing the payload. The excellent energy absorption capacity of cellular structures should not be overlooked either, as they can be used to create the necessary shock absorption zones during collisions, which absorb the energy of impacts in a designed manner, thus protecting the physical safety of operators [1,2,3,4]. Cellular structures should not be neglected in medical technology either, where their importance is also constantly growing. At the moment, such structures are widely used to improve osteointegration. Cellular structures are used for implants that are fixed without the use of adhesives. The stability of implants fixed by this method can be divided into two parts. The first is the so-called primary stability, which is provided by the geometry of the implant itself and is established immediately after surgery. Secondary stability is developed later—because it requires a process of ossification—during which the bone grows onto the implant and fixes it in place. The cellular structures are designed to promote secondary stability. The bone can grow into the structured surface and form mechanical fastenings, thereby increasing stability and reducing the risk of loosening. Nowadays, metal spraying and hydroxyapatite application are commonly used to ensure that the surface of implants is sufficiently structured. Furthermore, a surface structure is usually created by laser engraving, which also increases the degree of surface structure. On the other hand, we use a three-dimensional metal printed implant; the surface structure can be placed directly in the model, or it can be created by changing the printing parameters, since the surface roughness also changes with them (laser energy and laser spot size). Another huge advantage of implants made with three-dimensional metal printing is that the same surface modifications can be created on them as on implants made with traditional manufacturing technologies. However, with the increasing focus on patient-specific or at least partially customized implants, additive manufacturing technologies are emerging to structure the surface directly during the manufacturing process, thus reducing the number of production steps and simplifying the manufacturing process. Furthermore, by placing the cellular structure not only on the surface but also in the entire cross-section, it is possible to store a drug in implants with a programed release, allowing a high local concentration of the drug combined with a defined absorption time. Thus, one of the major advantages of adhesive implant fixation—namely programmed drug release—can be realized and further developed, as drug release in bone cement was not predictable and practically only feasible in cement fixation [5,6,7,8]. Furthermore, the so-called stress shielding phenomenon can be reduced by an implant with a cellular structure throughout its cross-section, as the cells can be used to bring the elastic modulus of the structure to a level equivalent to that of bone. This minimizes the micromovements at the bone–implant interface due to the very different mechanical properties of the implant. Furthermore, stress shielding is minimized—as equivalent mechanical properties promote adequate stress dissipation to the peri-implant areas—thus ensuring the best and most uniform load transfer between the bone and the implant [5,9]. This is important because the bone is a biologically economical structure and therefore tends to achieve maximum strength with minimum energy input and weight. For this reason, the structure of our bones is constantly changing in response to load, with the dynamic repair of damage occurring in the process. Thus, if the load transfer of the implant is not adequate and uniform, there will be parts of the bone that do not receive sufficient load stimuli, so the bone tissue, which is permanently adapted to the stresses, will fall victim to the processes of bone resorption, at the end of which the implant will loosen or break [10]. It is because of these processes that the phenomenon of stress shielding is so dangerous and must be avoided. Figure 1 shows some applications where cellular structures are already used in biomedical applications.

It can be seen that the application of cellular structures is nowadays mostly implemented in high-value, innovative, modern technological devices. With the right mechanical testing methods and simulation techniques, unnecessary production of very expensive products can be avoided, and considerable time and money can be saved [6,13]. For cellular structures, too, the development of appropriate mechanical and simulation testing methods is inevitable. These have been relatively little addressed to date and often with insufficient accuracy.

If we investigate the available literature, the inaccuracies mentioned earlier are also apparent in the measurements. In practically all the tests where displacement and, from that, strain had to be measured, the authors only calculated with the displacement of the crosshead of the universal material tester [1,2,13,14,15,16,17,18,19,20,21,22] which gives inaccurate results. Without proper fine strain measurement, the measured deformation will include the complete deformation of the load cell and the whole load chain (grips, adapters), including the indentation of the specimen into the pressure plates during loading. Experience has shown that this can result in false, several times higher measured deformations in the small deformation range at the high load being tested, leading to misleading material parameters. This is particularly true for compression tests. It could be said of virtually all the articles comparing FE (Finite Element) simulations with measured results that the measured stiffness or modulus results were much lower than the results of FE simulations, especially at higher loads [13,14,16,17,18], most likely because the results were calculated from crosshead displacement. For this reason, the ISO 13314:2011 standard [20] for mechanical testing of periodic cellular structures recommends the use of a strain gauge for displacement measurements [18]. Campoli [18] mentions in his article that they worked on the basis of this standard, but the reported results show that they did not apply the recommendation of the standard and calculated the deformation from crosshead displacement. In a paper by Gümrük [23], a deflectometer is used to measure displacement, which is considered to be slightly better than the calculation from crosshead displacement but still very inaccurate. Furthermore, in another paper by Gümrük and Smith also, the test specimens were coated with lubricant to reduce friction [16,17]. We would like to point out that the use of lubricant does not make much sense when testing cellular structures because the contact area is rather modest, and therefore the specimen will, in any case, press into the compression plates and cause an imprint. In the same way, a major problem in the Sallica-Leva [15] measurements is the penetration of specimens with too small an edge length into the compressive plates. To reduce this indentation, Herrera printed a solid block on the test specimen’s contact surfaces with the test plates [13]. However, this solution made it difficult to check the dimensional accuracy of the production since it was not possible to calculate the mass, and the application of µCT (Micro Computed Tomograph) was considerably complicated. However, the deviation between simulations and measured values still varied between 24.9 and 30.1% [13]. Among the articles examined, the only case in which the calculation from crosshead displacement is acceptable is that of Limmahakhun [21], because the photopolymer they use is destroyed at 1–2 MPa pressure. This is such a tiny load that it does not cause significant deformation of the testing equipment, but at the same time, this material is completely unsuitable as an implant material because of its softness. Based on measurements on sensitized knee implants at certain angular rotations, up to 4.7–7.6 × BW of body weight can be placed on an implant [5,24], so it is clear that an implant must be able to withstand a load greater than the 1–2 MPa pressure achieved in this article. Additionally, the photopolymer material used is prone to water absorption, as the authors have carried out measurements in both dried and wet conditions [21].

From a medical engineering point of view, several articles have investigated cellular structures, generally with the aim of achieving mechanical properties and proper osseointegration similar to human bone tissue and reducing stress shielding. The elastic modulus of the cortical bone cortex (corticalis) is 7–30 GPa, while the Young’s modulus of the spongiosa is 0.5–1.5 GPa [5,24].

Among the literature reviewed, there was also one that examined cellular structures from a manufacturing technology perspective. Yan and co-researchers found a relatively good geometric agreement between the designed CAD (Computer Aided Design) models and the fabricated structures. However, it was observed that the size of the finished products was slightly larger than the models, resulting in a decrease in pore size compared to the designed ones [1]. Smith et al. have also pointed out this phenomenon, as they found that the more filigree unit cells are used, the higher the strength [17].

It can be seen that previous studies presented in the introduction have investigated many aspects of the mechanical properties of cellular structures. Still, it can be said that despite many promising aspects, they are often not with sufficient care and precision. Therefore, our aim is to present an inspection and notation structure that can be applied universally and with sufficient accuracy to the categorization sizing of periodic cellular structures—in our case, highlighting medical applications. Our ultimate goal is to develop a test method for cellular structured implant systems with the same stiffness as bone tissue, with the help of which the mechanical sizing and personalization of artificial joints can be carried out in everyday medical practice.

## 2. Materials and Methods

### 2.1. Selection of Cell Structure

From the literature reviewed, it was found that in many cases, there was a problem with the exact definition of the cell structure—For example, in Maghoddam’s article [25], we could only deduce the cell structure based on the image, because it was completely omitted—so we have prepared a short summary of the most commonly used structures. This can be seen in detail in Figure 2.

Figure 2 shows the open-cell structures that have been studied and used in medical technology and in the literature. In addition to these periodic cellular structures, there are countless other structures, such as bio-inspired structures (bamboo and sheep’s horn) [24]. We have tried to show all common cell structures with unit cells and the polar diagram showing the direction dependency of stiffness obtained from the FE simulation. In the FE simulations, the RVE (Representative Volume Element) unit cells of every structure were classified according to the von Mises criterion. In the preliminary FE simulations, the structures were meshed with 1-dimensional elements, and then, for clarification, the FE simulations were also performed with a 3-dimensional mesh. Before the FE simulation tests, we performed the relevant convergence test and only performed the simulations after the convergence criterion was met. However, due to space constraints, we have only included the polar diagram if the anisotropy was appropriate for medical uses or if it had been used in previous research. The selection of the cell structure we investigated began with a simulation to determine which structure properties were least direction-dependent. Then, we examined the structure for osteointegration and programmed release drug storage. By looking at the polar diagrams and comparing the numerical simulation data, it is clear that the best structures in terms of anisotropy are the so-called TPMS (Triply Periodic Minimal Surfaces) structures, among which is the Walled TPMS SplitP. The difference between the TPMS and Walled TPMS cell structures is that in the case of Walled TPMS the wall thickness of the resulting structure can be freely modulated, which is why a W is added to the boxes of the TPMS cell structures (Figure 2). Modulation of the wall thickness is very advantageous, as it allows the pore size to be varied for a given cell size, thus achieving the optimal pore size for osseointegration. However, structures with different beam elements are favorable for high porosity, which is important because of the larger volume of the active substance left for drug storage. Furthermore, this allows for the storage of drugs with a programed release, which dissolve over different time periods and also occupy a significant volume. It can be seen that of the three requirements, anisotropy and osseointegration are contradicted by drug storage. However, the Walled TPMS SplitP fully satisfies the requirements of both osseointegration and anisotropy, and this structure was chosen for our investigations.

It is recommended to follow the presented procedure for specifying the structure, as this allows us to define the structure used in a clear, precise, and simple way. This requires the cell type, the cell diameter, wall thickness, or beam diameter—depending on whether it is a beam or walled TPMS or other structure—and then the porosity.

Simulations and specimen generation were carried out in nTopology (version: 3.36.3, nTop Inc., New York, NY, USA) software, which has modules supporting both additive manufacturing technology and mechanical and thermal simulations.

For the demonstration of proper characterization, we produced two cell sizes and wall thicknesses, as we wanted to see how the structure would behave at the two extremes and what the manufacturing and simulation problems would be. Naturally, we tested the structures in the intermediate values by simulations. One of the cellular structures had a wall thickness of 2 mm, while the cell diameter was 14 mm; this was the coarse structure with a high wall thickness, while the fine structure had a wall thickness of 0.4 mm and a cell diameter of 3.4 mm. Furthermore, it was an important consideration that the mechanical properties of the two selected specimens matched those of the human bone tissue, which was set by preliminary simulations. Figure 3 shows two characteristic 30 × 30 × 30 mm specimens.

### 2.2. Production of Specimens

The production was done with DMLS (Direct Metal Laser Sintering) technology using an EOS M100 printer (EOS GmbH, Kralling, Germany) with the EOS company parameter set EOS_DirectPart (engraving speed 1400 mm/s, LASER power 100 W, and 0.02 mm layer thickness), the material used was standard, medical grade EOS Ti_6_Al_4_V (EOS GmbH, Kralling, Germany). Autodesk Netfabb Premium 2022 software (version: 2022.0, Autodesk Inc., San Rafael, CA, USA) was used to debug the models, and Materialize Magics 24 (Materialise NV, Leuven, Belgium) was used for slicing.

### 2.3. Test Methods

The compression tests were carried out on a Zwick Z250 universal materials testing machine (Zwick Roell, Ulm, Germany). The strain was measured using a Mercury Monet DIC (Digital Image Correlation)-based optical measurement system (Mercury MS, s.r.o, Brno, Czech Republic). The resolution of the used monochrome camera was 5 MPixel. Figure 4 shows the measurement setup and a specimen with pre-painted pattern during measurement. The test speed was 2 mm/min. The tests were carried out at room temperature.

## 3. Results and Discussion

Since our goal was to develop an analysis method that can be used universally for cellular structures, we summarized the steps and tried to point out the mistakes a user should avoid during the process.

In the first step, we generated specimens with different porosities—varying both the cell size and the wall thickness—and simulated the elastic modulus of the structures. The results were plotted in a diagram (Figure 5).

This is important because it provides information on how the structure responds to changes in porosity, and it is also useful for extrapolation later on using the resulting curve. This is very important because it is possible to infer elastic modulus or porosity values without having to run time-consuming and expensive simulations or measurements.

The investigations were continued with compressive testing of the printed samples. A problem during the measurements was that these complex cellular structures behave differently from solid specimens during measurements. The cells are not just compressed but are constantly being rearranged. This means that the cells can rotate and slip on each other. This phenomenon was most pronounced at the beginning of the measurements, at low load levels, where the degree of deformation was comparable to the degree of cell aligning.

For this very reason, we had to perform multiple iterations to obtain a suitable measurement from which we could calculate an elastic modulus. We repeatedly increased the applied force, in which the final measurement was 10,000 N. During our measurements, we examined the location of the probes, placed in a grid in several locations on the specimen for improved accuracy. In particular, we tried to place the measurement probes at the boundaries of the cells or at given distances from each other. The positioning of the probes is shown in Figure 6. Figure 7 shows the measurement and simulation results of the fine cell structure.

Figure 6 shows how we tried to use as many probes as possible in our measurements and averaged the results. One of the significant advantages of DIC measurements is that they can be used to collect a virtually infinite amount of data from a single measurement, unlike, for example, strain gauge arrangements.

Figure 7 shows that the modulus values obtained vary depending on how far apart the sampling points are placed in the DIC measurement.

From the measurements, it was immediately apparent that the modulus values obtained were much higher than the results previously expected from the simulations. To solve this problem, we had to turn to the literature, as several articles mentioned that the printed specimens were slightly larger than previously designed models [1,2]. This was the case here, as the mass measurements showed that the resulting specimens were much heavier than calculated from the CAD model. Examining the sample under an optical microscope and electron microscope, we found that there were many areas in the pores where the material powder had melted and adhered to the surface, even though the LASER was not scanning that area. Measurements taken with an optical microscope (Figure 8) showed that this process did not really affect the cell dimensions but rather thickened the cell walls by reducing the pores.

Based on what we learned from the microscopic images, we ran new simulations where we did not change the cell size but increased the cell wall thickness. Autodesk Fusion 360 is an additional option for determining various deviations, which can help in identifying errors; the program is used in countless areas, for example, even for the analysis of injection molded spur gears [26]. From the mass, we calculated the corresponding porosity value, for which we can get a good approximation with the extrapolation curve, but in our case, we performed the exact simulation. The corrected simulation showed a very good correlation with the measured values (Figure 7).

Therefore, it is crucial to perform a corrected simulation after measuring the printed test specimen’s manufacturing inaccuracy. Furthermore, the initially designed structure should be chosen to be softer than the desired modulus value.

As a next step, the nTopology simulation was validated by an Ansys simulation (version: 2021 R2, Ansys Inc., Canonsburg, PA, USA), by running a simulation of the original test piece. In nTopology, it is possible to export a finite element mesh that can be analyzed in Ansys, but nTopology does not export a model that Ansys can mesh properly. ANSYS is an established and highly reliable simulation software, which is used for countless things from thermal simulation to vibration and noise simulation [27], so it can be used to qualify nTopology simulations. In the diagram (Figure 7), we observed that we obtained practically the same results with the two programs. It is important to point out here that both nTopology and Ansys allow the selection of the elements used. As shown in the literature, the use of beam (1 dimensional) elements will result in a relatively fast calculation with low computational requirements but with inaccurate results. Furthermore, beam elements can only be used if the structure is built of beam elements. The use of a three-dimensional solid model is significantly slower and more computationally demanding but gives more accurate results [16,28,29]. It is mostly applicable to TPMS specimens. So, for approximate preliminary calculations, the beam element is acceptable, but for more accurate calculations, the solid model is necessary.

In the case of one cell, it can be seen that the variance of the result is substantial, although it is within the corrected simulation range. This can be explained by the fact that it is unfortunate to have too short sampling lengths and that cells are also aligning (rotating and shifting) during the measurement. With more cells, the effect of rotation and drift is less pronounced, so it is recommended to include more cell sampling lengths. However, care should also be taken not to get too close to the contacts of the compressive plates, where the transverse movement of the specimen is increasingly restricted. This is also reflected in the measurement results, as the variance of the results obtained at 23 mm gauge length is very small, and the diagram clearly shows that the structure becomes softer as the test piece approaches the test plate. Therefore, it can be concluded from the measurement results that a gauge length of at least two cells is required for sufficiently accurate measurement and that the sampling points should not be closer than two cells to the footprint. If these parameters are kept and the longest possible gauge length is used, a reliable and realistic result can be obtained.

After reviewing the relevant literature, it was found that practically all research groups have so far calculated elastic modulus values from the crosshead displacement of the universal material testing machine, but Figure 7 shows that this is not practical, as it would also include the mechanical properties of the machine in the results. It can be seen that the value calculated from the crosshead displacement is practically an order of magnitude smaller than the values measured with the DIC and the simulated values. The difference between the stress-displacement curves calculated from the crosshead displacement and the stress-displacement curves calculated from the optical strain measurement is shown in Figure 9. It can be clearly seen that the slope of the curve calculated from the crosshead displacement is much smaller, and also an initial settling region is present; therefore, the calculated strain values are much higher than those calculated from the DIC measurement and the FE simulated values, resulting in lower evaluated moduli values.

This is particularly true in cases where a smaller specimen contact area is used or a higher load is applied, as the material testing equipment will deform more due to the higher surface pressure. Deformation is best observed on the test specimen support plates, as shown in Figure 10.

Figure 10 shows a photograph and optical microscope (Keyence VHX 5000—Keyence Corporation, Osaka, Japan) image of the compressive plate imprint, clearly showing the bottom layer of the cell structure penetrating the surface causing irreversible deformation. Several measurements have been taken, and therefore several structures can be observed. In addition, we used a GOM Atos Core 5M three-dimensional scanner (Gesellschaft für Optische Messtechnik GmbH, Carl Zeiss AG, Jena, Germany) to create a point cloud of the imprint, thus quantifying the penetration depth. A plane was superimposed on the surface of the point cloud and used by the software (GOM Inspect 2018—version 113294, Gesellschaft für Optische Messtechnik GmbH, Carl Zeiss AG, Jena, Germany) to plot the deviations similar to the contour lines on a map. The warm colors indicate the positive deviations from the reference surface, while the cold colors indicate the negative deviations from the reference surface, and the green indicates the original surface. It can be seen that the indentation is in the hundredths of a millimeter range—and the contour of the test specimen is visible on the test plate—so this has a major influence on the measurement, as the penetration is comparable to the deformation of the test specimen.

Based on the lessons learned from the previous tests, it is more difficult to test the coarser cell structure since it is impossible to keep two cells at a distance from the plates and still have a gauge length of 2 cells. However, we definitely wanted to test such a specimen, as there may be cases where the above requirement cannot be met, possibly due to the size, geometry, or cell structure of the component. However, it is also necessary to measure in such a situation, and we wanted to find a solution. Having carried out the steps detailed previously, we obtained the result presented in Figure 11.

After mass measurement and optical microscopy, it was found that the geometric dimensions of the test specimen were practically identical to the CAD model, so no corrected simulation was necessary. This is also in line with the literature, which states that the larger the cell size, the better the accuracy [1]. This can logically be explained by the fact that the areas to be melted are further apart, so the surrounding powder receives much less heat, and the heat input is much more localized, so less powder particles will be undesirably melted.

We also ran the control simulation in Ansys software (version: 2021 R2, Ansys Inc., Canonsburg, PA, USA), which in this case did not match the nTopology simulation as closely but showed more correlation with the measurement results. It is clear that the variance of Young’s modulus values obtained from the Ansys simulation is also orders of magnitude larger than in the previous case, so the anisotropy of the test sample and the uncertainty of the calculation are larger. This is most likely the case in this situation, as there is no significant difference between the values, but the variance is seriously increased.

The situation with the measurement is similar to that with the simulation, as the uncertainty is much larger than in the previous case. It can be seen that the variance is high for the 23 mm measurement length, which can most probably be explained by the fact that the inhibition due to the pressure conditions is also present. This was manifested in the fact that the cellular structure could be twisted at the measurement point, thus increasing the uncertainty. However, it is clearly observed that it makes sense to place the measurement points in a well-identifiable location, such as the pore edges, as there are far fewer points that can be tracked by DIC on this test. Obviously, this makes it challenging to achieve the correct measurement length, but the accuracy will be much higher than if the software cannot track the measurement point. In a number of cases, the software’s inability to track a measurement point in a geometrically difficult-to-identify location has been a problem.

In order to determine the anisotropy and possible drug storage, simulations were carried out by cutting a cube and a sphere from a cubic specimen and investigating the elastic modulus of this structure. Figure 12 shows the sphere and cube cut from the test specimen.

The cube cutout was selected because it exactly follows the geometry of the original specimen, while the sphere is a rounded body, allowing us to model how much the inner cutout needs to be rounded to avoid anisotropic behavior and how many rows of cells can be cut out. The simulation results of the hollow structures are shown in Figure 13.

As can be seen in Figure 13, the elastic modulus decreases, and the anisotropy (difference in minimum and maximum modulus values) increases with the amount of material cut out. It is clear that to maintain isotropy, a sphere—i.e., some rounded geometry—should be chosen. In our opinion, the structure is unacceptable if the deviation between the simulated maximum and minimum modulus values, indicating anisotropy, is greater than 10%. Thus, it can be seen that for a non-rounded cutout, a minimum of 1.25 rows of cells are required to remain—since the test specimen contained five cells in total—and for a rounded cutout, the required remaining cell volume is half a row. If we round these to the nearest whole number, we obtain that two rows of cells are required to maintain isotropy in the case of an unrounded cutout and one row in the case of a rounded cutout. This simulation is also important because it shows that if we take a minimum gauge length of two cells, we will definitely have an anisotropic structure.

Since the requirements of drug storage contradict the requirements of osseointegration and desired isotropy, we prepared a simulation test specimen with a TPMS structure on the outside and a Kelvin cell structure on the inside as a further research option (Figure 14). This allows for isotropy and good osseointegration using the TPMS structure, as well as high internal porosity due to the Kelvin cell structure composed of beam elements. The Kelvin cell structure was chosen because it is one of the most isotropic of the beam element cells at the same time having excellent porosity. The resulting hybrid cell structure component effectively acts as a coupled spring system, so the two cell structures support each other, meaning that an even larger volume can be cut out of the TPMS shell and filled with programmed release drugs, and the material suspension and mass are reduced. As with such hybrid systems, the potentially localized anisotropy outlined by Deering et al. for the individual and specific situation can be achieved [28].

## 4. Conclusions

The literature review revealed that the fabrication and testing of periodic cell structures are still in a very rudimentary form, often rather imprecise and unsystematic. Therefore, our aim was to develop a universally applicable testing method to accurately tune the mechanical properties of periodic cell structures.

Firstly, we recommend following our procedure for specifying the structure, which requires specifying the cell type, cell diameter, wall thickness, or beam diameter—depending on whether it is a beam or walled TPMS (Triply Periodic Minimal Surface) or other structure—and then the porosity. In this way, the periodic cellular structure used can be clearly and accurately identified.

Secondly, the following technique can be used to tune the stiffness of the structure to specific needs. The sequence of steps of the test method developed:Choosing the cell structure (depending on the desired purpose, the polar diagram is helpful).Preparing an extrapolation diagram (using FE simulations—Finite Element).Designing the test piece in CAD (Computer Aided Design) software (choosing a smaller modulus of elasticity and larger pore size for manufacturing inaccuracies—reducing the number of costly and energy-intensive iterations).Performing a control measurement to compensate for the manufacturing inaccuracies and then using this data to run a control simulation (mass and microscopic measurement of pores or beam diameter to check the dimensions of the finished specimens—if the values obtained from the control simulation are not acceptable, a further iteration is required).Carrying out the mechanical tests with the following recommendations:Measurement of displacement using optical contactless strain measurement.Generation a sufficiently large displacement (sufficient deflection for proper strain measurement).Recording the longest possible measurement length:Minimum measuring length of two cell diameters.Maintain a minimum distance of two cell diameters from the contact surfaces for measurement points.Measurement points should be placed at geometrically well-identifiable locations.


With the help of the measurement method we developed, an accuracy of 5–8% can be achieved in the case of the first iteration. However, if we run the sizing method several times, this accuracy can be significantly increased, since the number of iterations is increased. It can be seen that by applying the exact selection and sizing method we created, the results are comparable and more reliable than in existing studies, regardless of cellular structure [1,2,13,14,15,16,17,18,19,20,21,22].

From the point of view of medical applications, we can define other possible proposals for which our measurement methodology can be adapted. One is the use of different hybrid cell structures, which allows the storage of programmed-release drugs and the design of fully customized local isotropies. The other is the investigation of load transfer of fully customized prostheses at the bone-implant interface based on CT (Computer Tomography) images.

## Figures and Tables

**Figure 1 materials-16-04146-f001:**
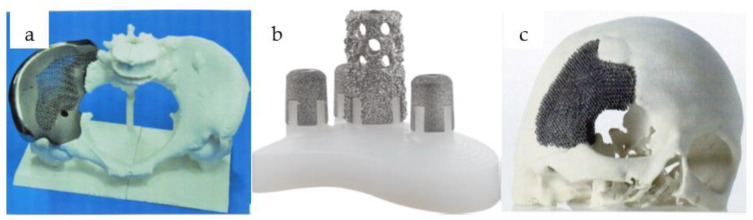
Application of cellular structures: partial pelvic bone replacement (**a**) reprinted from [11], permission from Elsevier, license number: 5521760059478, glenoid fixation of shoulder implant (**b**) reprinted from [12], permission from Elsevier, license number: 5521771381185, partial skull replacement (**c**) reprinted from [11], permission from Elsevier, license number: 5521760059478.

**Figure 2 materials-16-04146-f002:**
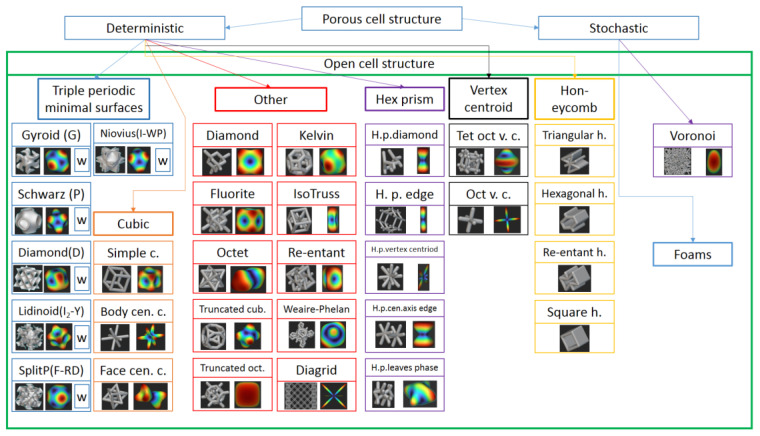
Summary and classification of cellular structures. (The colors represent the different groups of cellular structures. In the case of TPMS structures, the letters written in parentheses after the name represent the abbreviation of the given structure, and the W refers to the possibility of the Walled TPMS structure for the given structure).

**Figure 3 materials-16-04146-f003:**
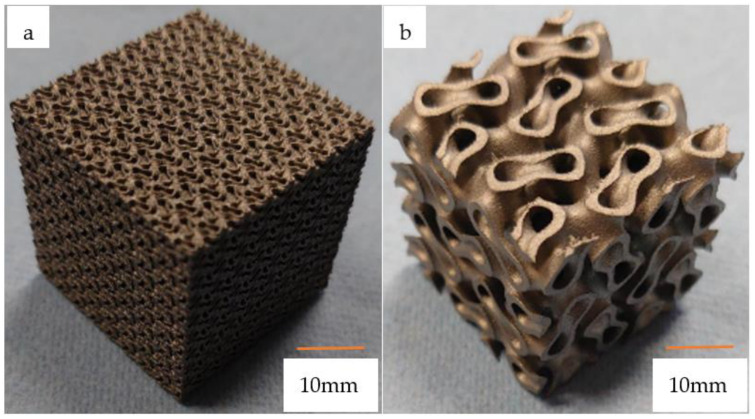
The two specimens ((**a**). The left one with a cell diameter of 3.4 mm and a wall thickness of 0.4 mm, (**b**). the right one with a cell diameter of 14 mm and a wall thickness of 2 mm).

**Figure 4 materials-16-04146-f004:**
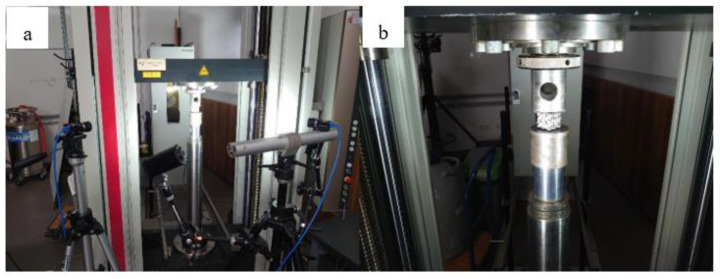
The measurement setup (**a**), and a test specimen with random pattern (**b**).

**Figure 5 materials-16-04146-f005:**
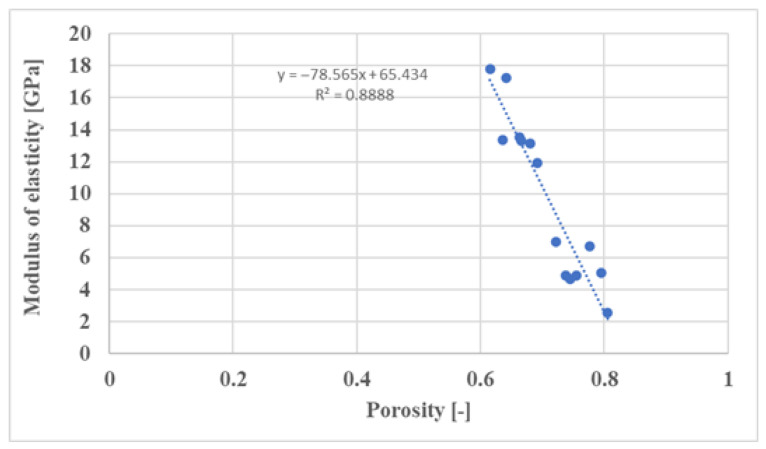
Extrapolation diagram for the Walled TPMS SplitP cell structure. (It can be seen that, depending on the porosity, the modulus changes practically linearly based on the learning of the straight line fitted to the simulation points.).

**Figure 6 materials-16-04146-f006:**
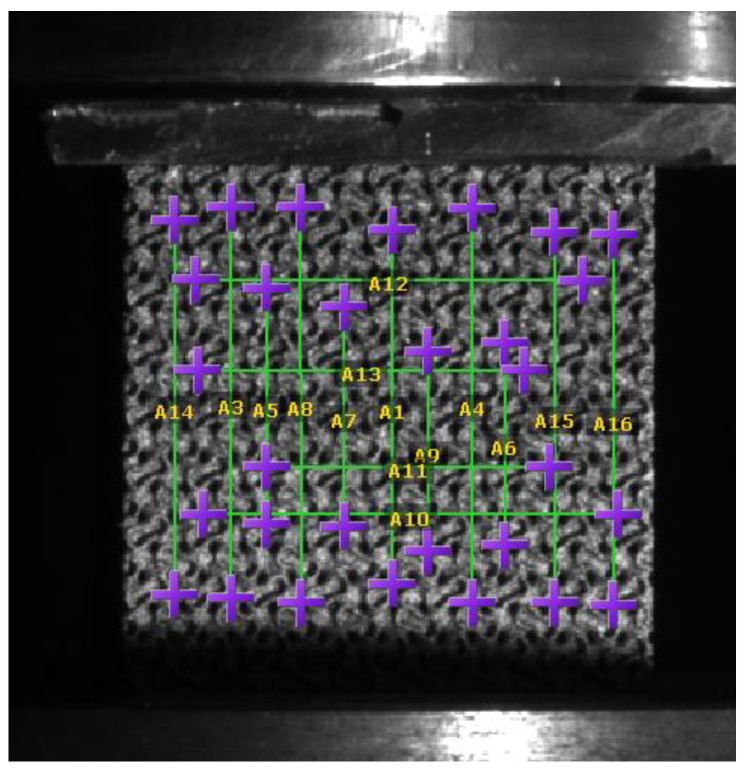
Placement of test probes. (The probes are constructed in such a way that the purple crosses are the end points, and the green lines connecting them indicate the measuring lengths between the corresponding crosses.)

**Figure 7 materials-16-04146-f007:**
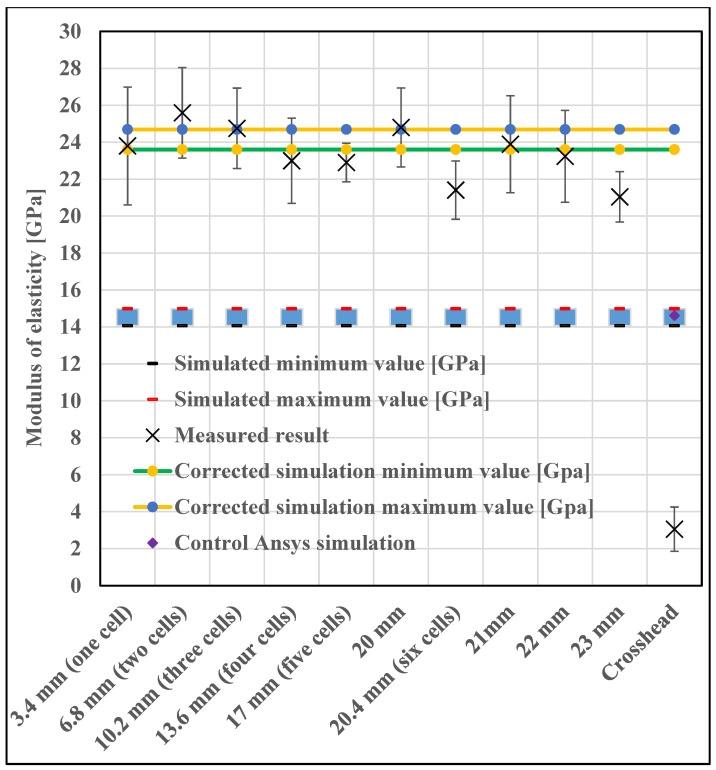
Comparison of measurement and simulation results for a fine structure cube (cell diameter 3.4 mm, wall thickness 0.4 mm, the blue squares indicate the modulus between the maximum and minimum simulation values).

**Figure 8 materials-16-04146-f008:**
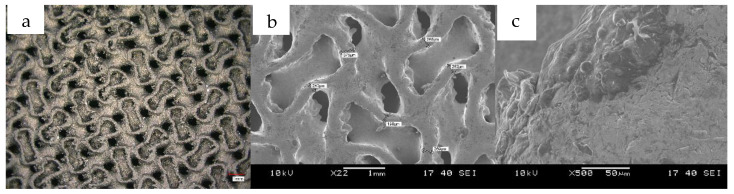
Optical microscope (**a**) and SEM image (**b**) of the surface of the specimen, the 500× magnification SEM image (**c**) clearly shows the excess material adhered to the walls during printing.

**Figure 9 materials-16-04146-f009:**
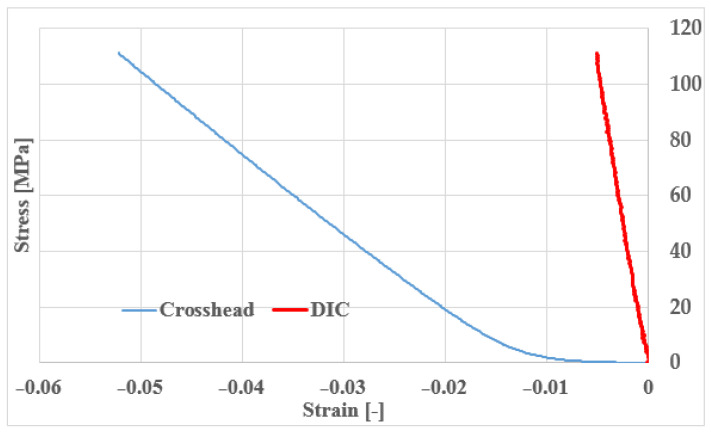
Difference between the stress–strain curve calculated from crosshead displacement and optical strain (DIC—digital image correlation) measurement.

**Figure 10 materials-16-04146-f010:**
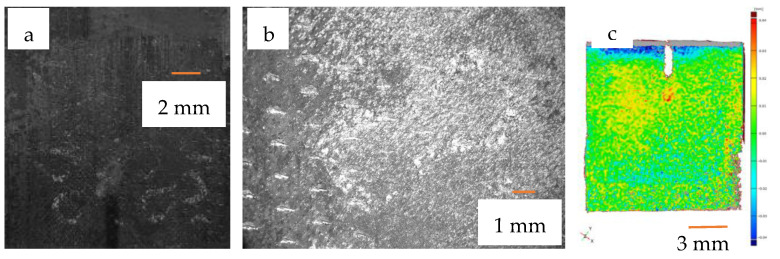
Photograph (**a**), optical microscope image (**b**) and 3D scanned image (**c**) of the footprint after examination.

**Figure 11 materials-16-04146-f011:**
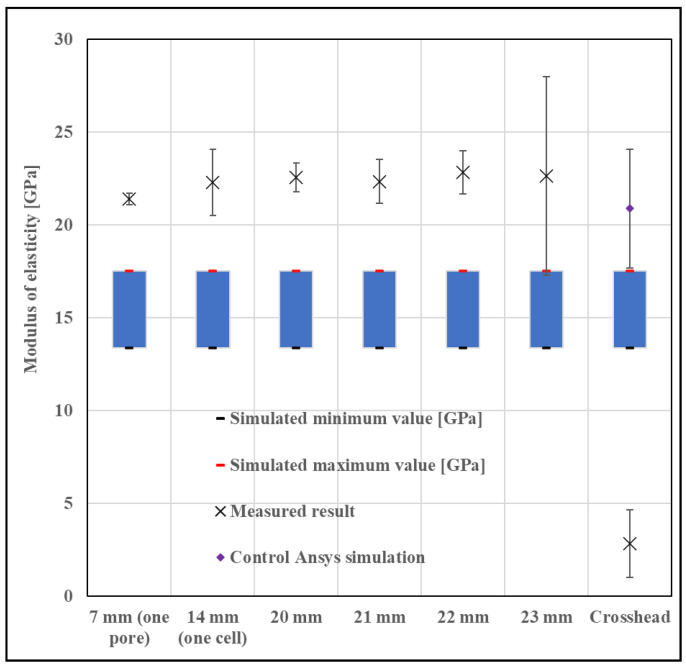
Comparison of measurement and simulation results for a coarse cube (cell diameter 14 mm, wall thickness 2 mm, the blue squares indicate the modulus between the maximum and minimum simulation values).

**Figure 12 materials-16-04146-f012:**
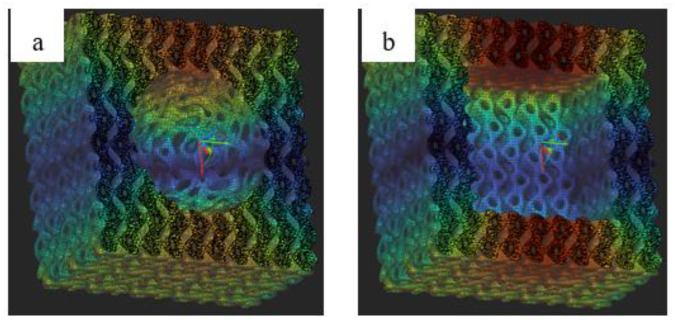
The rounded (spherical) shape cut from the test specimen (**a**) and the cubic cut following the original geometry of the test specimen (**b**) (Shades represent the stress distribution in the figure, as these are FE simulation specimens.).

**Figure 13 materials-16-04146-f013:**
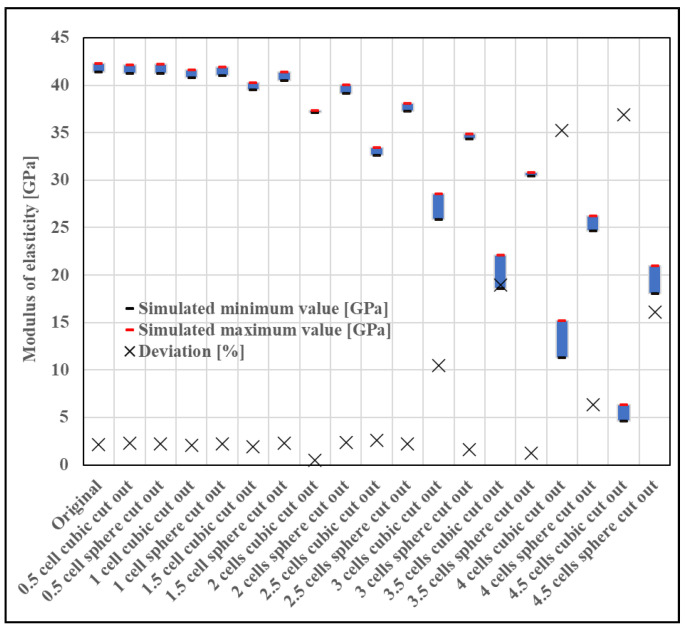
Result of an anisotropy test performed by cutting out a cube or sphere (The blue squares indicate the modulus between the maximum and minimum simulation values.).

**Figure 14 materials-16-04146-f014:**
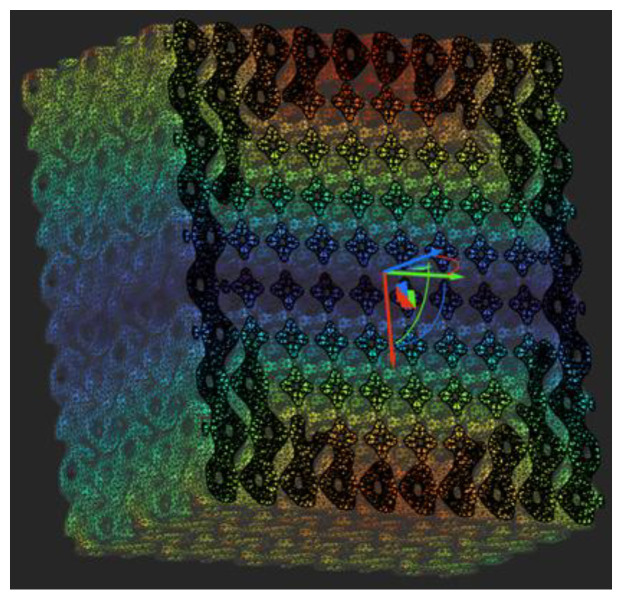
Hybrid cell structure (Shades represent the stress distribution in the figure, as these are FE simulation specimens.).

## Data Availability

The data presented in this study are available on request from the corresponding author.

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
