# Peer review of "Reliable Methods for Classification, Characterization, and Design of Cellular Structures for Patient-Specific Implants"

_materials, 2023, doi:10.3390/ma16114146_

Round 1
Reviewer 1 Report
The authors proposed a characterization method that can be universally applied to periodic cell structures, focusing on medical applications. This work has certain engineering application significance. However, there are certain points that need attention, and I recommend that the paper be accepted for publication once it is revised in accordance with the following comments:
(1) The innovative point in the introduction is not clearly stated and the framework of the article is not explained;
(2) The quality of the image in Figure 2 is low;
(3) Figure 3 lacks a scale;
(4) What are the literatures referred to on lines 299 and 321?
(5) In fact, it is not "clearly" from Figure 9 that the honeycomb structure penetrates the surface, resulting in an irreversible deformation.
Author Response
Dear Reviewer,
Firstly, the authors would like to thank the Reviewers for Your thorough and comprehensive corrections and their help to create a better, more valuable article worthy of publication. In the enclosed document we give point-by-point answers to your remarks. Please also have a look at the revised manuscript to see the evolution of the article.
In the hope that this manuscript will be accepted in this revised form.
We wish you, your family and colleagues good health and safety.
Sincerely yours,
Dr. Gábor Szebényi
associate professor, Head of Material Testing Laboratory
BME Department of Polymer Engineering

Reviewer 2 Report
The topic is relevant to current context but needs to incorporate all the modifications suggested in the below mentioned form-
|
● Before submission of the manuscript do check the instructions for author, abstract should be less than 200 words but it has already crossed currently showing 231 words. Abstract underscores the content as there is no connectivity to the reader why this work is being carried out? The work currently look like a review article rather than a research article. Significance of the work? Need of the work has to be explained in the early part of abstract ● There is no statistical information as an outcome achieved in this work, which is a must for abstract. ● As per journal guidelines up to 10 keywords can be preferred, still there is scope for addition (6 keywords mentioned) ● References should be mentioned as per the journal guidelines ● Authors have to restrict the self-citations. Which looks to be not a good practice |
|
· There is still lot of scope for improvising the introduction section as it talks limited in terms of the current trends in combination of FE simulations and mechanical compression tests with fine strain measurement. There are plenty of recent articles to support few of them have been listed in below. You are free to select other relevant articles- ● doi: 10.3390/biomimetics7040186 ● doi.org/10.1016/j.aej.2021.03.074 ● 10.1007/s00894-020-04560-9 There is neither any discussion on convergence criteria nor on the comparison data of predecessor work
|
|
● How this cellular structure came to consideration, was it a bio-inspired or was its AI/ML based or something else has to be clearly mentioned in the background ● Figure 2 image is not clear and no one could able to judge mere base on image. Its mandatory that work has to be explained with case studies. ● Nowhere any information about the FE simulation software, mentioned as RVE. Readers should be able to mimic the work for any future cases so it’s necessary to provide detailed information ● Figure 3 shows two combinations of models developed but was is the base for it. Is the dimension based on any earlier benchmarked work or it’s new? ● Lot many parameters considered as specifications directly without prior base. Why only this specification? Justify ● There were no simulation studies carried out or even experimental to arrive at the outcome of these two models as shown in figure 3. ● The designed structures seem to be good but there is no justification for the same. ● There is no information about of the elemental mesh size, elemental type, order of the elements. No clarity on the no. of elements or nodes in the whole 2D meshed model ● The whole simulation work could have been better shown in the form of pictorial representation or a process map |
|
● From figure 4 it can be seen that only compression tests carried out but as implant will have buckling as well as impact load acting on the component. There is no information about the same. ● Figure 5 talks about modulus of elasticity is it statistically generated or experimentally arrived? ● What is the base for 10000N load consideration? ● There are no images pertaining to mesh generated, contact generated, load and boundary condition applied in FE simulation? Is the similar aspects such as loads and boundary conditions considered as in real-time testing? ● Figure 7, what exactly you are trying to discuss? Is it higher amount of material being printed or what? In what way this will help the structure? ● Results and discussions are acceptable only with any correlation between the current used material and existing materials in public domain ● Authors are advised to look into the molecular dynamics results once again to recalculate the variation in figure 10. For better understanding refer the following article- ● 10.1016/B978-0-12-819904-6.00022-0 ● There is no comparative study with analytical/experimental/simulation for validation of the extracted results ● There is no information about convergence study or convergence criteria selected for this FEA simulation |
|
● The discussion section lags in explanation with respect to the work carried out. there are no citations in discussion section to compare the work with existing materials ● Conclusion looks to be generic need to compile the outcomes and state based on the tests conducted and convey how best this can fit in the current context for any application. ● In conclusion section, values have to be displayed with explanation. It's better to mention the salient features of the entire work in terms of bullet points with current context |

Native English speaker has to look into the manuscript.
Author Response

(The authors gave the same response as above.)

Reviewer 3 Report
In the review of the manuscript entitled Reliable methods for classification, characterization, and design of cellular structures for patient-specific implants. The authors have provided a good description and the methodology is also fine. I would like to see this article publish but after some questions as follow;
1. What was the goal of the research?
2. What is the importance of accurate tuning of the stiffness properties of cellular structure components in medical applications?
3. How can porous, cellular structures reduce the need for revision surgeries?
4. How can stress shielding and micromovements at the bone-implant interface be reduced?
5. What is the significance of storing drugs with a programmed mechanism of action inside implants with a cellular structure?
6. What is the current state of literature on stiffness sizing procedure for periodic cellular structures?
7. What is the proposed uniform marking system for cellular structures?
8. What is the multi-step exact stiffness design and validation methodology developed in the research?
9. What techniques were used in the validation methodology?
10. How accurate is the stiffness setting achieved through the developed methodology?
Author Response

(The authors gave the same response as above.)

Reviewer 4 Report
The following issues and questions should be addressed:
1. I would suggest to avoid the usage of abbreviations in the abstract (FE; line 22). If an abbreviation is used, it should be defined the first time it is used. Also here: line 103.
2. It should be indicated that the authors have the allowance to use the images from other journals in the figure caption.
3. Figure 2 looks very blurry and the images are very small. This figure must be improved. When you zoom in, you can see individual pixels, which means that the quality of this images needs to be increased. Moreover, the size of the individual images used in this figures are too small. Also the fonts used is particularly different (style, size, …). Please unify the font.
4. Line 202: the cubic is missing for the mm here. Moreover, instead of the letter x, the symbol × (U+2A2F) should be used.
5. Figure 3: Better to add a scale bar, to improve the readability of the manuscript.
6. Throughout the whole manuscript, abbreviations were used without defining them before the first usage. For example: TPMS; line 174.
7. Which software was used to model structures with mechanical properties of bone tissue (lines 200 – 202). Specify the name, software version, company, city, country of the manufacturer.
8. Which mechanical properties of the bone tissue were taken into account, it is necessary to indicate the physical quantities and their numerical values.
9. It is also necessary to provide additional information (city, country of manufacture) for the 3D printer indicated in the article and Ti powder. What dispersion powder was used, you also need to indicate the brand of powder (lines 210 – 213).
9. What software was used to slice the product before printing? Specify the name, city, country of manufacturer and software version.
10. Line 215, the device Zwick Z250 is not indicated correctly. All used devices in this study should be indicated correctly: (type, company, city, country).
11. Based on figures 3, 6, etc., the gyroid structure was chosen. However, the motivation for choosing this particular filling pattern is not explained properly. Please add a suitable explanation in the introduction, supported by references.
12. Figure 4 is useless, since it is showing only a universal testing device. I would used it only in an additionally support information manuscript. In this way, it is just blowing up the manuscript.
13. Figure 6 is too small and a scale bar is missing.
14. Figure 7: The scale bars are completely unsuitable and should be bigger and improved.
15. Figure 8: The right y-axis labelling is in the diagram. Please correct this issue. DIC abbreviation should be defined in the figure caption.
16. Figure 9: A scale bar for each image is missing. Please add them.
17. Figure 11 and 13: Scale bars are missing. Please add them.
18. For a good conclusion, abbreviation should be avoided.
19. Impersonal expressions are encouraged in a scientific manuscript (pronouns such as "we" and "us" should be avoided).
20. The structure of the manuscript should be increased. Sometimes a figure is mentioned in a paragraph and not before or after the paragraph placed. This manuscript is containing 13 figures where some figures can be combined to better structure the manuscript. For example: Figure 11 and Figure 13.
21. What about the surface roughness and wettability of the samples shown in figure 3, since these are important parameters for cell adhesion? Can these parameters also be tuned for the needs of different cell types?
22. The infill density is not mentioned for the samples used in this study. It should be also discussed why this grade of infill density has been used in this study. What are the advantages and disadvantages for a lower and higher infill density on the stiffness propertied and tunability?
23. For the introduction, the authors should also mention the differences between metal implants, biostable polymeric implants and biodegradable polymeric implants. Moreover, the need for surface modification of implants via magnetron sputtering, microarc-oxidation, different etching methods or biocompatible coatings and films should be also added into the introduction.
No comments.
Author Response

(The authors gave the same response as above.)

Round 2
Reviewer 2 Report
Authors have successfully addressed the comments and the manuscript can be accepted in the present form.
Minor English correction has to be made on the manuscript.
Reviewer 4 Report
The authors of the manuscript entitled “Reliable methods for classification, characterization, and design of cellular structures for patient-specific implants” addressed and discussed the most of my comments in an appropriate way. After revision, the quality of the manuscript raised significantly. I have nothing more to add. Therefore, I would like to suggest the editor to accept this manuscript.
No comments.